# Incremental Clustering: The Case for Extra Clusters

**Margareta Ackerman**
Florida State University
600 W College Ave, Tallahassee, FL 32306
mackerman@fsu.edu

**Sanjoy Dasgupta**
UC San Diego
9500 Gilman Dr, La Jolla, CA 92093
dasgupta@eng.ucsd.edu

## Abstract

The explosion in the amount of data available for analysis often necessitates a transition from batch to *incremental* clustering methods, which process one element at a time and typically store only a small subset of the data. In this paper, we initiate the formal analysis of incremental clustering methods focusing on the types of cluster structure that they are able to detect. We find that the incremental setting is strictly weaker than the batch model, proving that a fundamental class of cluster structures that can readily be detected in the batch setting is impossible to identify using any incremental method. Furthermore, we show how the limitations of incremental clustering can be overcome by allowing additional clusters.

## 1 Introduction

Clustering is a fundamental form of data analysis that is applied in a wide variety of domains, from astronomy to zoology. With the radical increase in the amount of data collected in recent years, the use of clustering has expanded even further, to applications such as personalization and targeted advertising. Clustering is now a core component of interactive systems that collect information on millions of users on a daily basis. It is becoming impractical to store all relevant information in memory at the same time, often necessitating the transition to *incremental* methods.

Incremental methods receive data elements one at a time and typically use much less space than is needed to store the complete data set. This presents a particularly interesting challenge for unsupervised learning, which unlike its supervised counterpart, also suffers from an absence of a unique target truth. Observe that not all data possesses a meaningful clustering, and when an inherent structure exists, it need not be unique (see Figure 1 for an example). As such, different users may be interested in very different partitions. Consequently, different clustering methods detect distinct types of structure, often yielding radically different results on the same data. Until now, differences in the input-output behaviour of clustering methods have only been studied in the batch setting [12, 13, 8, 4, 3, 5, 2, 19]. In this work, we take a first look at the types of cluster structures that can be discovered by incremental clustering methods.

To qualify the type of cluster structure present in data, a number of notions of *clusterability* have been proposed (for a detailed discussion, see [1] and [8]). These notions capture the structure of the *target clustering*: the clustering desired by the user for a specific application. As such, notions of clusterability facilitate the analysis of clustering methods by making it possible to formally ascertain whether an algorithm correctly recovers the desired partition.

One elegant notion of clusterability, introduced by Balcan et al. [8], requires that every element be closer to data in its own cluster than to other points. For simplicity, we will refer to clusterings that adhere to this requirement as *nice*. It was shown by [8] that such clusterings are readily detected offline by classical batch algorithms. On the other hand, we prove (Theorem 3.8) that no incremental method can discover these partitions. Thus, batch algorithms are significantly stronger than incremental methods in their ability to detect cluster structure.

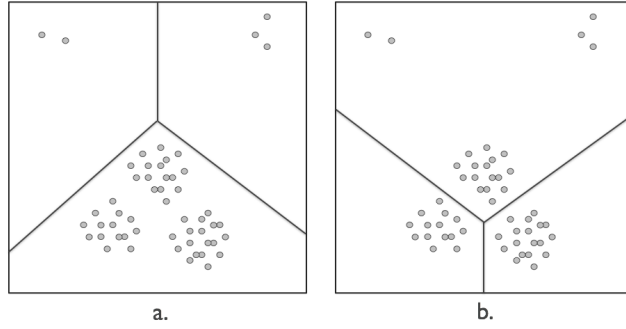

Figure 1: An example of different cluster structures in the same data. The clustering on the left finds inherent structure in the data by identifying well-separated partitions, while the clustering on the right discovers structure in the data by focusing on the dense region. The correct partitioning depends on the application at hand.

In an effort to identify types of cluster structure that incremental methods can recover, we turn to stricter notions of clusterability. A notion used by Epter et al. [9] requires that the minimum separation between clusters be larger than the maximum cluster diameter. We call such clusterings *perfect*, and we present an incremental method that is able to recover them (Theorem 4.3).

Yet, this result alone is unsatisfactory. If, indeed, it were necessary to resort to such strict notions of clusterability, then incremental methods would have limited utility. Is there some other way to circumvent the limitations of incremental techniques?

It turns out that incremental methods become a lot more powerful when we slightly alter the clustering problem: if, instead of asking for exactly the target partition, we are satisfied with a *refinement*, that is, a partition each of whose clusters is contained within some target cluster. Indeed, in many applications, it is reasonable to allow additional clusters.

Incremental methods benefit from additional clusters in several ways. First, we exhibit an algorithm that is able to capture nice $k$-clusterings if it is allowed to return a refinement with $2^{k-1}$ clusters (Theorem 5.3), which could be reasonable for small $k$. We also show that this exponential dependence on $k$ is unavoidable in general (Theorem 5.4). As such, allowing additional clusters enables incremental techniques to overcome their inability to detect nice partitions.

A similar phenomenon is observed in the analysis of the sequential $k$-means algorithm, one of the most popular methods of incremental clustering. We show that it is unable to detect perfect clusterings (Theorem 4.4), but that if each cluster contains a significant fraction of the data, then it can recover a refinement of (a slight variant of) nice clusterings (Theorem 5.6).

Lastly, we demonstrate the power of additional clusters by relaxing the niceness condition, requiring only that clusters have a significant *core* (defined in Section 5.3). Under this milder requirement, we show that a randomized incremental method is able to discover a refinement of the target partition (Theorem 5.10).

Due to space limitations, many proofs appear in the supplementary material.

## 2 Definitions

We consider a space $\mathcal{X}$ equipped with a symmetric distance function $d : \mathcal{X} \times \mathcal{X} \to \mathbb{R}^+$ satisfying $d(x, x) = 0$. An example is $\mathcal{X} = \mathbb{R}^p$ with $d(x, x') = \|x - x'\|_2$. It is assumed that a clustering algorithm can invoke $d(\cdot, \cdot)$ on any pair $x, x' \in \mathcal{X}$.

A *clustering* (or, *partition*) of $\mathcal{X}$ is a set of clusters $\mathcal{C} = \{C_1, \dots, C_k\}$ such that $C_i \cap C_j = \emptyset$ for all $i \neq j$, and $\mathcal{X} = \cup_{i=1}^k C_i$. A *k-clustering* is a clustering with $k$ clusters.

Write $x \sim_\mathcal{C} y$ if $x, y$ are both in some cluster $C_j$; and $x \not\sim_\mathcal{C} y$ otherwise. This is an equivalence relation.

**Definition 2.1.** *An* **incremental clustering algorithm** *has the following structure:*

*for $n = 1, \ldots, N$:*
  *See data point $x_n \in \mathcal{X}$*
  *Select model $M_n \in \mathcal{M}$*

*where $N$ might be $\infty$, and $\mathcal{M}$ is a collection of clusterings of $\mathcal{X}$. We require the algorithm to have bounded memory, typically a function of the number of clusters. As a result, an incremental algorithm cannot store all data points.*

Notice that the ordering of the points is unspecified. In our results, we consider two types of ordering: *arbitrary ordering*, which is the standard setting in online learning and allows points to be ordered by an adversary, and *random ordering*, which is standard in statistical learning theory. In *exemplar-based clustering*, $\mathcal{M} = \mathcal{X}^k$: each model is a list of $k$ "centers" $(t_1, \ldots, t_k)$ that *induce* a clustering of $\mathcal{X}$, where every $x \in \mathcal{X}$ is assigned to the cluster $C_i$ for which $d(x, t_i)$ is smallest (breaking ties by picking the smallest $i$). All the clusterings we will consider in this paper will be specified in this manner.

We also note that the incremental clustering model is closely related to streaming clustering [6, 10], the primary difference being that in the latter framework multiple passes of the data are allowed.

## 2.1 Examples of incremental clustering algorithms

The most well-known incremental clustering algorithm is probably *sequential $k$-means*, which is meant for data in Euclidean space. It is an incremental variant of Lloyd's algorithm [16, 17]:

**Algorithm 2.2.** *Sequential $k$-means.*

*Set $T = (t_1, \ldots, t_k)$ to the first $k$ data points*
*Initialize the counts $n_1, n_2, ..., n_k$ to $1$*
*Repeat:*
  *Acquire the next example, $x$*
  *If $t_i$ is the closest center to $x$:*
    *Increment $n_i$*
    *Replace $t_i$ by $t_i + (1/n_i)(x - t_i)$*

This method, and many variants of it, have been studied intensively in the literature on self-organizing maps [15]. It attempts to find centers $T$ that optimize the $k$-means cost function:

$$\text{cost}(T) = \sum_{\text{data } x} \min_{t \in T} \|x - t\|^2.$$

It is not hard to see that the solution obtained by sequential $k$-means at any given time can have cost far from optimal; we will see an even stronger lower bound in Theorem 4.4. Nonetheless, we will also see that if additional centers are allowed, this algorithm is able to correctly capture some fundamental types of cluster structure.

Another family of clustering algorithms with incremental variants are *agglomerative* procedures [12] like single-linkage [11]. Given $n$ data points in batch mode, these algorithms produce a hierarchical clustering on all $n$ points. But the hierarchy can be truncated at the intermediate $k$-clustering, yielding a tree with $k$ leaves. Moreover, there is a natural scheme for updating these leaves incrementally:

**Algorithm 2.3.** *Sequential agglomerative clustering.*

*Set $T$ to the first $k$ data points*
*Repeat:*
  *Get the next point $x$ and add it to $T$*
  *Select $t, t' \in T$ for which* `dist`$(t, t')$ *is smallest*
  *Replace $t, t'$ by the single center* `merge`$(t, t')$

Here the two functions `dist` and `merge` can be varied to optimize different clustering criteria, and often require storing additional sufficient statistics, such as counts of individual clusters. For instance, Ward's method of average linkage [18] is geared towards the $k$-means cost function. We will consider the variant obtained by setting `dist`$(t, t') = d(t, t')$ and `merge`$(t, t')$ to either $t$ or $t'$:

**Algorithm 2.4.** *Sequential nearest-neighbour clustering.*

*Set $T$ to the first $k$ data points*
*Repeat:*
　　*Get the next point $x$ and add it to $T$*
　　*Let $t, t'$ be the two closest points in $T$*
　　*Replace $t, t'$ by either of these two points*

We will see that this algorithm is effective at picking out a large class of cluster structures.

## 2.2  The target clustering

Unlike supervised learning tasks, which are typically endowed with a unique correct classification, clustering is ambiguous. One approach to disambiguating clustering is identifying an objective function such as $k$-means, and then defining the clustering task as finding the partition with minimum cost. Although there are situations to which this approach is well-suited, many clustering applications do not inherently lend themselves to any specific objective function. As such, while objective functions play an essential role in deriving clustering methods, they do not circumvent the ambiguous nature of clustering.

The term *target clustering* denotes the partition that a specific user is looking for in a data set. This notion was used by Balcan et al. [8] to study what constraints on cluster structure make them efficiently identifiable in a batch setting. In this paper, we consider families of target clusterings that satisfy different properties, and ask whether incremental algorithms can identify such clusterings.

The target clustering $\mathcal{C}$ is defined on a possibly infinite space $\mathcal{X}$, from which the learner receives a sequence of points. At any time $n$, the learner has seen $n$ data points and has some clustering that ideally agrees with $\mathcal{C}$ on these points. The methods we consider are *exemplar-based*: they all specify a list of points $T$ in $\mathcal{X}$ that induce a clustering of $\mathcal{X}$ (recall the discussion just before Section 2.1). We consider two requirements:

- (Strong) $T$ induces the target clustering $\mathcal{C}$.
- (Weaker) $T$ induces a refinement of the target clustering $\mathcal{C}$: that is, each cluster induced by $T$ is part of some cluster of $\mathcal{C}$.

If the learning algorithm is run on a finite data set, then we require these conditions to hold once all points have been seen. In our positive results, we will also consider infinite streams of data, and show that these conditions hold at every time $n$, taking the target clustering restricted to the points seen so far.

## 3  A basic limitation of incremental clustering

We begin by studying limitations of incremental clustering compared with the batch setting.

One of the most fundamental types of cluster structure is what we shall call *nice clusterings* for the sake of brevity. Originally introduced by Balcan et al. [8] under the name "strict separation," this notion has since been applied in [2], [1], and [7], to name a few.

**Definition 3.1** (Nice clustering). *A clustering $\mathcal{C}$ of $(\mathcal{X}, d)$ is* **nice** *if for all $x, y, z \in \mathcal{X}$, $d(y, x) < d(z, x)$ whenever $x \sim_{\mathcal{C}} y$ and $x \not\sim_{\mathcal{C}} z$.*

See Figure 2 for an example.

**Observation 3.2.** *If we select one point from every cluster of a nice clustering $\mathcal{C}$, the resulting set induces $\mathcal{C}$. (Moreover, niceness is the minimal property under which this holds.)*

A nice $k$-clustering is not, in general, unique. For example, consider $\mathcal{X} = \{1, 2, 4, 5\}$ on the real line under the usual distance metric; then both $\{\{1\}, \{2\}, \{4, 5\}\}$ and $\{\{1, 2\}, \{4\}, \{5\}\}$ are nice 3-clusterings of $\mathcal{X}$. Thus we start by considering data with a *unique* nice $k$-clustering.

Since niceness is a strong requirement, we might expect that it is easy to detect. Indeed, in the batch setting, a unique nice $k$-clustering can be recovered by single-linkage [8]. However, we show that nice partitions cannot be detected in the incremental setting, even if they are unique.

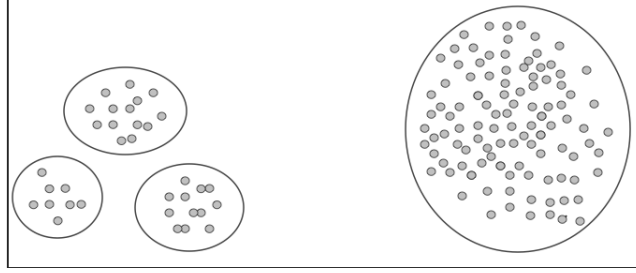

Figure 2: A nice clustering may include clusters with very different diameters, as long as the distance between any two clusters scales as the larger diameter of the two.

We start by formalizing the ordering of the data. An *ordering function* $O$ takes a finite set $\mathcal{X}$ and returns an ordering of the points in this set. An *ordered distance space* is denoted by $(O[\mathcal{X}], d)$.

**Definition 3.3.** *An incremental clustering algorithm $\mathcal{A}$ is **nice-detecting** if, given a positive integer $k$ and $(\mathcal{X}, d)$ that has a unique nice $k$-clustering $\mathcal{C}$, the procedure $\mathcal{A}(O[\mathcal{X}], d, k)$ outputs $\mathcal{C}$ for any ordering function $O$.*

In this section, we show (Theorem 3.8) that no deterministic memory-bounded incremental method is nice-detecting, even for points in Euclidean space under the $\ell_2$ metric.

We start with the intuition behind the proof. Fix any incremental clustering algorithm and set the number of clusters to 3. We will specify a data set $D$ with a unique nice 3-clustering that this algorithm cannot detect. The data set has two subsets, $D_1$ and $D_2$, that are far away from each other but are otherwise nearly isomorphic. The target 3-clustering is either: ($D_1$, together with a 2-clustering of $D_2$) or ($D_2$, together with a 2-clustering of $D_1$).

The central piece of the construction is the configuration of $D_1$ (and likewise, $D_2$). The first point presented to the learner is $x_o$. This is followed by a clique of points $x_i$ that are equidistant from each other and have the same, slightly larger, distance to $x_o$. For instance, we could set distances within the clique $d(x_i, x_j)$ to 1, and distances $d(x_i, x_o)$ to 2. Finally there is a point $x'$ that is *either* exactly like one of the $x_i$'s (same distances), *or* differs from them in just one specific distance $d(x', x_j)$ which is set to 2. In the former case, there is a nice 2-clustering of $D_1$, in which one cluster is $x_o$ and the other cluster is everything else. In the latter case, there is no nice 2-clustering, just the 1-clustering consisting of all of $D_1$.

$D_2$ is like $D_1$, but is rigged so that if $D_1$ has a nice 2-clustering, then $D_2$ does not; and vice versa.

The two possibilities for $D_1$ are almost identical, and it would seem that the only way an algorithm can distinguish between them is by remembering all the points it has seen. A memory-bounded incremental learner does not have this luxury. Formalizing this argument requires some care; we cannot, for instance, assume that the learner is using its memory to store individual points.

In order to specify $D_1$, we start with a larger collection of points that we call an *$M$-configuration*, and that is independent of any algorithm. We then pick two possibilities for $D_1$ (one with a nice 2-clustering and one without) from this collection, based on the specific learner.

**Definition 3.4.** *In any metric space $(\mathcal{X}, d)$, for any integer $M > 0$, define an $M$-configuration to be a collection of $2M + 1$ points $x_o, x_1, \ldots, x_M, x_1', \ldots, x_M' \in \mathcal{X}$ such that*

- *All interpoint distances are in the range $[1, 2]$.*

- $d(x_o, x_i), d(x_o, x_i') \in (3/2, 2]$ *for all $i \geq 1$.*

- $d(x_i, x_j), d(x_i', x_j'), d(x_i, x_j') \in [1, 3/2]$ *for all $i \neq j \geq 1$.*

- $d(x_i, x_i') > d(x_o, x_i).$

The significance of this point configuration is as follows.

**Lemma 3.5.** *Let $x_o, x_1, \ldots, x_M, x'_1, \ldots, x'_M$ be any $M$-configuration in $(\mathcal{X}, d)$. Pick any index $1 \le j \le M$ and any subset $S \subset [M]$ with $|S| > 1$. Then the set $A = \{x_o, x'_j\} \cup \{x_i : i \in S\}$ has a nice 2-clustering if and only if $j \notin S$.*

*Proof.* Suppose $A$ has a nice 2-clustering $\{C_1, C_2\}$, where $C_1$ is the cluster that contains $x_o$.

We first show that $C_1$ is a singleton cluster. If $C_1$ also contains some $x_\ell$, then it must contain all the points $\{x_i : i \in S\}$ by niceness since $d(x_\ell, x_i) \le 3/2 < d(x_\ell, x_o)$. Since $|S| > 1$, these points include some $x_i$ with $i \ne j$. Whereupon $C_1$ must also contain $x'_j$, since $d(x_i, x'_j) \le 3/2 < d(x_i, x_o)$. But this means $C_2$ is empty.

Likewise, if $C_1$ contains $x'_j$, then it also contains all $\{x_i : i \in S, i \ne j\}$, since $d(x_i, x'_j) < d(x_o, x'_j)$. There is at least one such $x_i$, and we revert to the previous case.

Therefore $C_1 = \{x_o\}$ and, as a result, $C_2 = \{x_i : i \in S\} \cup \{x'_j\}$. This 2-clustering is nice if and only if $d(x_o, x'_j) > d(x_i, x'_j)$ and $d(x_o, x_i) > d(x'_j, x_i)$ for all $i \in S$, which in turn is true if and only if $j \notin S$. $\qquad \square$

By putting together two $M$-configurations, we obtain:

**Theorem 3.6.** *Let $(\mathcal{X}, d)$ be any metric space that contains two $M$-configurations separated by a distance of at least 4. Then, there is no deterministic incremental algorithm with $\le M/2$ bits of storage that is guaranteed to recover nice 3-clusterings of data sets drawn from $\mathcal{X}$, even when limited to instances in which such clusterings are unique.*

*Proof.* Suppose the deterministic incremental learner has a memory capacity of $b$ bits. We will refer to the memory contents of the learner as its *state*, $\sigma \in \{0, 1\}^b$.

Call the two $M$-configurations $x_o, x_1, \ldots, x_M, x'_1, \ldots, x'_M$ and $z_o, z_1, \ldots, z_M, z'_1, \ldots, z'_M$. We feed the following points to the learner:

> Batch 1:     $x_o$ and $z_o$
> Batch 2:     $b$ distinct points from $x_1, \ldots, x_M$
> Batch 3:     $b$ distinct points from $z_1, \ldots, z_M$
> Batch 4:     Two final points $x'_{j_1}$ and $z'_{j_2}$

The learner's state after seeing batch 2 can be described by a function $f : \{x_1, \ldots, x_M\}^b \to \{0, 1\}^b$. The number of distinct sets of $b$ points in batch 2 is $\binom{M}{b} > (M/b)^b$. If $M \ge 2b$, this is $> 2^b$, which means that two different sets of points must lead to the same state, call it $\sigma \in \{0, 1\}^b$. Let the indices of these sets be $S_1, S_2 \subset [M]$ (so $|S_1| = |S_2| = b$), and pick any $j_1 \in S_1 \setminus S_2$.

Next, suppose the learner is in state $\sigma$ and is then given batch 3. We can capture its state at the end of this batch by a function $g : \{z_1, \ldots, z_M\}^b \to \{0, 1\}^b$, and once again there must be distinct sets $T_1, T_2 \subset [M]$ that yield the same state $\sigma'$. Pick any $j_2 \in T_1 \setminus T_2$.

It follows that the sequences of inputs $x_o, z_o, (x_i : i \in S_1), (z_i : i \in T_2), x'_{j_1}, z'_{j_2}$ and $x_o, z_o, (x_i : i \in S_2), (z_i : i \in T_1), x'_{j_1}, z'_{j_2}$ produce the same final state and thus the same answer. But in the first case, by Lemma 3.5, the unique nice 3-clustering keeps the $x$'s together and splits the $z$'s, whereas in the second case, it splits the $x$'s and keeps the $z$'s together. $\qquad \square$

An $M$-configuration can be realized in Euclidean space:

**Lemma 3.7.** *There is an absolute constant $c_o$ such that for any dimension $p$, the Euclidean space $\mathbb{R}^p$, with $L_2$ norm, contains $M$-configurations for all $M < 2^{c_o p}$.*

The overall conclusions are the following.

**Theorem 3.8.** *There is no memory-bounded deterministic nice-detecting incremental clustering algorithm that works in arbitrary metric spaces. For data in $\mathbb{R}^p$ under the $\ell_2$ metric, there is no deterministic nice-detecting incremental clustering algorithm using less than $2^{c_o p - 1}$ bits of memory.*

# 4   A more restricted class of clusterings

The discovery that nice clusterings cannot be detected using any incremental method, even though they are readily detected in a batch setting, speaks to the substantial limitations of incremental algorithms. We next ask whether there is a well-behaved subclass of nice clusterings that can be detected using incremental methods. Following [9, 2, 5, 1], among others, we consider clusterings in which the maximum cluster diameter is smaller than the minimum inter-cluster separation.

**Definition 4.1** (Perfect clustering). *A clustering $\mathcal{C}$ of $(\mathcal{X}, d)$ is* **perfect** *if $d(x, y) < d(w, z)$ whenever $x \sim_\mathcal{C} y$, $w \not\sim_\mathcal{C} z$.*

Any perfect clustering is nice. But unlike nice clusterings, perfect clusterings are unique:

**Lemma 4.2.** *For any $(\mathcal{X}, d)$ and $k$, there is at most one perfect $k$-clustering of $(\mathcal{X}, d)$.*

Whenever an algorithm can detect perfect clusterings, we call it *perfect-detecting*. Formally, an incremental clustering algorithm $\mathcal{A}$ is ***perfect-detecting*** if, given a positive integer $k$ and $(\mathcal{X}, d)$ that has a perfect $k$-clustering, $\mathcal{A}(O[\mathcal{X}], d, k)$ outputs that clustering for any ordering function $O$.

We start with an example of a simple perfect-detecting algorithm.

**Theorem 4.3.** *Sequential nearest-neighbour clustering (Algorithm 2.4) is perfect-detecting.*

We next turn to sequential $k$-means (Algorithm 2.2), one of the most popular methods for incremental clustering. Interestingly, it is unable to detect perfect clusterings.

It is not hard to see that a perfect $k$-clustering is a local optimum of $k$-means. We will now see an example in which the perfect $k$-clustering is the global optimum of the $k$-means cost function, and yet sequential $k$-means fails to detect it.

**Theorem 4.4.** *There is a set of four points in $\mathbb{R}^3$ with a perfect 2-clustering that is also the global optimum of the $k$-means cost function (for $k = 2$). However, there is no ordering of these points that will enable this clustering to be detected by sequential $k$-means.*

# 5   Incremental clustering with extra clusters

Returning to the basic lower bound of Theorem 3.8, it turns out that a slight shift in perspective greatly improves the capabilities of incremental methods. Instead of aiming to exactly discover the target partition, it is sufficient in some applications to merely uncover a refinement of it. Formally, a clustering $\mathcal{C}$ of $\mathcal{X}$ is a *refinement* of clustering $\mathcal{C}'$ of $\mathcal{X}$, if $x \sim_\mathcal{C} y$ implies $x \sim_{\mathcal{C}'} y$ for all $x, y \in \mathcal{X}$.

We start by showing that although incremental algorithms cannot detect nice $k$-clusterings, they can find a refinement of such a clustering if allowed $2^{k-1}$ centers. We also show that this is tight.

Next, we explore the utility of additional clusters for sequential $k$-means. We show that for a random ordering of the data, and with extra centers, this algorithm can recover (a slight variant of) nice clusterings. We also show that the random ordering is necessary for such a result.

Finally, we prove that additional clusters extend the utility of incremental methods beyond nice clusterings. We introduce a weaker constraint on cluster structure, requiring only that each cluster possess a significant "core", and we present a scheme that works under this weaker requirement.

## 5.1   An incremental algorithm can find nice $k$-clusterings if allowed $2^k$ centers

Earlier work [8] has shown that that any nice clustering corresponds to a pruning of the tree obtained by single linkage on the points. With this insight, we develop an incremental algorithm that maintains $2^{k-1}$ centers that are guaranteed to induce a refinement of any nice $k$-clustering.

The following subroutine takes any finite $S \subset \mathcal{X}$ and returns at most $2^{k-1}$ distinct points:

CANDIDATES($S$)
    Run single linkage on $S$ to get a tree
    Assign each leaf node the corresponding data point
    Moving bottom-up, assign each internal node the data point in one of its children
    Return all points at distance $< k$ from the root

**Lemma 5.1.** *Suppose $S$ has a nice $\ell$-clustering, for $\ell \leq k$. Then the points returned by* CANDIDATES*$(S)$ include at least one representative from each of these clusters.*

Here's an incremental algorithm that uses $2^{k-1}$ centers to detect a nice $k$-clustering.

**Algorithm 5.2.** *Incremental clustering with extra centers.*

$T_0 = \emptyset$
*For $t = 1, 2, \ldots$:*
    *Receive $x_t$ and set $T_t = T_{t-1} \cup \{x_t\}$*
    *If $|T_t| > 2^{k-1}$: $T_t \leftarrow$* CANDIDATES$(T_t)$

**Theorem 5.3.** *Suppose there is a nice $k$-clustering $\mathcal{C}$ of $\mathcal{X}$. Then for each $t$, the set $T_t$ has at most $2^{k-1}$ points, including at least one representative from each $C_i$ for which $C_i \cap \{x_1, \ldots, x_t\} \neq \emptyset$.*

It is not possible in general to use fewer centers.

**Theorem 5.4.** *Pick any incremental clustering algorithm that maintains a list of $\ell$ centers that are guaranteed to be consistent with a target nice $k$-clustering. Then $\ell \geq 2^{k-1}$.*

## 5.2 Sequential $k$-means with extra clusters

Theorem 4.4 above shows severe limitations of sequential $k$-means. The good news is that additional clusters allow this algorithm to find a variant of nice partitionings.

The following condition imposes structure on the convex hull of the partitions in the target clustering.

**Definition 5.5.** *A clustering $\mathcal{C} = \{C_1, \ldots, C_k\}$ is **convex-nice** if for any $i \neq j$, any points $x, y$ in the convex hull of $C_i$, and any point $z$ in the convex hull of $C_j$, we have $d(y, x) < d(z, x)$.*

**Theorem 5.6.** *Fix a data set $(\mathcal{X}, d)$ with a convex-nice clustering $\mathcal{C} = \{C_1, \ldots, C_k\}$ and let $\beta = \min_i |C_i|/|\mathcal{X}|$. If the points are ordered uniformly at random, then for any $\ell \geq k$, sequential $\ell$-means will return a refinement of $\mathcal{C}$ with probability at least $1 - ke^{-\beta\ell}$.*

The probability of failure is small when the refinement contains $\ell = \Omega((\log k)/\beta)$ centers. We can also show that this positive result no longer holds when data is adversarially ordered.

**Theorem 5.7.** *Pick any $k \geq 3$. Consider any data set $\mathcal{X}$ in $\mathbb{R}$ (under the usual metric) that has a convex-nice $k$-clustering $\mathcal{C} = \{C_1, \ldots, C_k\}$. Then there exists an ordering of $\mathcal{X}$ under which sequential $\ell$-means with $\ell \leq \min_i |C_i|$ centers fails to return a refinement of $\mathcal{C}$.*

## 5.3 A broader class of clusterings

We conclude by considering a substantial generalization of niceness that can be detected by incremental methods when extra centers are allowed.

**Definition 5.8** (Core). *For any clustering $\mathcal{C} = \{C_1, \ldots, C_k\}$ of $(\mathcal{X}, d)$, the **core** of cluster $C_i$ is the maximal subset $C_i^o \subset C_i$ such that $d(x, z) < d(x, y)$ for all $x \in C_i$, $z \in C_i^o$, and $y \notin C_i$.*

In a nice clustering, the core of any cluster is the entire cluster. We now require only that each core contain a significant fraction of points, and we show that the following simple sampling routine will find a refinement of the target clustering, even if the points are ordered adversarially.

**Algorithm 5.9.** *Algorithm subsample.*

*Set $T$ to the first $\ell$ elements*
*For $t = \ell + 1, \ell + 2, \ldots$:*
  *Get a new point $x_t$*
  *With probability $\ell/t$:*
    *Remove an element from $T$ uniformly at random and add $x_t$ to $T$*

It is well-known (see, for instance, [14]) that at any time $t$, the set $T$ consists of $\ell$ elements chosen at random without replacement from $\{x_1, \ldots, x_t\}$.

**Theorem 5.10.** *Consider any clustering $\mathcal{C} = \{C_1, \ldots, C_k\}$ of $(\mathcal{X}, d)$, with core $\{C_1^o, \ldots, C_k^o\}$. Let $\beta = \min_i |C_i^o|/|\mathcal{X}|$. Fix any $\ell \geq k$. Then, given any ordering of $\mathcal{X}$, Algorithm 5.9 detects a refinement of $\mathcal{C}$ with probability $1 - ke^{-\beta\ell}$.*

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
