[Supplementary Material]

# APPENDIX
# Incremental Clustering: The Case for Extra Clusters

**Margareta Ackerman**
Florida State University
600 W College Ave, Tallahassee, FL 32306
mackerman@fsu.edu

**Sanjoy Dasgupta**
UC San Diego
9500 Gilman Dr, La Jolla, CA 92093
dasgupta@eng.ucsd.edu

## A basic limitation of incremental clustering

An $M$-configuration can be realized in Euclidean space:

**Lemma 3.7** There is an absolute constant $c_o$ such that for any dimension $p$, the Euclidean space $\mathbb{R}^p$, with $L_2$ norm, contains $M$-configurations for all $M < 2^{c_o p}$.

We will use the probabilistic method to construct an $M$-configuration in $\mathbb{R}^p$.

- Let $x_o$ be any vector of length $1 < a < 2$ (we will fix $a$ later).
- Pick $x_1, \ldots, x_M$ uniformly at random from the surface of the unit ball in $\mathbb{R}^p$.
- Set each $x'_i = -x_i$.

We will show that with probability $> 0$, the resulting set of points is an $M$-configuration; therefore, an $M$-configuration must exist.

We start by considering distances between $x_o$ and any other point.

**Lemma 3.8** Fix any $x_o \in \mathbb{R}^p$ of length $a$ and pick $X$ uniformly at random from the unit sphere in $\mathbb{R}^p$. Then $\mathbb{E}\|X - x_o\|^2 = a^2 + 1$, and for any $0 \le t \le 1$,

$$\Pr(|\|X - x_o\|^2 - (a^2 + 1)| > t) \ \le \ 2\exp(-t^2 p/(8a^2)).$$

*Proof.* First observe that

$$\|X - x_o\|^2 \ = \ \|X\|^2 + \|x_o\|^2 - 2X \cdot x_o \ = \ a^2 + 1 - 2X \cdot x_o.$$

When $X$ is chosen uniformly at random from the unit sphere, $\mathbb{E}(X \cdot x_o) = (\mathbb{E}X) \cdot x_o = 0$ and thus $\mathbb{E}\|X - x_o\|^2 = a^2 + 1$.

Next, define $f(x) = x \cdot x_o$. This function is $a$-Lipschitz with respect to the $\ell_2$ norm: for any $x, y \in \mathbb{R}^p$,

$$|f(x) - f(y)| \ = \ |x \cdot x_o - y \cdot x_o| \ \le \ \|x - y\|\|x_o\| \ = \ a\|x - y\|.$$

It follows by measure concentration on the unit sphere (see, for instance, Theorem 14.3.2 of [2]) that for $0 \le t \le 1$,

$$\Pr(|f(X) - \mathrm{med}(f)| > t) \le 2\exp(-t^2 p/(2a^2)).$$

Here $\mathrm{med}(f)$ is the median value of $f(X)$, which is $0$ by the symmetry of the distribution. Therefore,

$$\Pr(|\|X - x_o\|^2 - (a^2 + 1)| > t) \ = \ \Pr(|f(X)| > t/2) \ \le \ 2\exp(-t^2 p/(8a^2)),$$

as claimed. $\qquad\square$

## A more restricted class of clusterings

**Lemma 4.3** Sequential nearest-neighbour clustering is perfect-detecting.

*Proof.* Consider a data set that has a perfect $k$-clustering $\mathcal{C}$. We prove that the following invariant holds at any time in the execution of the algorithm: the clustering induced by the centers is a refinement of $\mathcal{C}$ restricted to the data seen so far.

Clearly, the above holds after the first $k$ elements are given, since each is made into a center. After the initial $k$ points are shown, every new point given to the algorithm becomes a center, and then the two closest centers are merged. We will now show that any two merged centers belong to the same cluster of $\mathcal{C}$; thus the invariant holds always.

Recall that in a perfect clustering, all within-cluster distances are strictly smaller than all between-cluster distances. Thus, of the $k+1$ centers, the two that merge (the two closest) must belong to the same cluster.

As soon as points are seen from all clusters of $\mathcal{C}$, the centers maintained by the algorithm induce $\mathcal{C}$. $\qquad\square$

**Theorem 4.4** There is a set of four points in $\mathbb{R}^3$ with a perfect 2-clustering that is also the global optimum of the $k$-means cost function (for $k = 2$). However, there is no ordering of these points that will enable this clustering to be detected by sequential $k$-means.

*Proof.* Consider these four points in $\mathbb{R}^3$: for $0 < \epsilon < 1/2$,

$$x_1 = (1, 0, 0), \quad x_2 = (-1, 0, 0) \quad x_3 = (0, 1, \sqrt{2} + \epsilon) \quad x_4 = (0, -1, \sqrt{2} + \epsilon).$$

These points have a perfect 2-clustering $\mathcal{C} = \{\{x_1, x_2\}, \{x_3, x_4\}\}$, with within-cluster distance 2, and between-cluster distances greater than 2. Moreover, this is also the global optimum of the $k$-means cost function, as can be checked by enumerating the various cases. However, we will now see that there is no ordering of the points that would enable this clustering to be detected by sequential $k$-means.

Suppose the first two points to be seen belong to the same cluster in $\mathcal{C}$. Then it can be checked that the next two points will get assigned to the same center, and will lead to a final clustering in which three of the points are grouped together. Thus assume, without loss of generality, that the first two points are $x_1$ and $x_3$, and again without loss of generality, that the next point is $x_2$.

Then, after the seeing the first three points, the cluster representatives are at $(0, 0, 0)$ and $(0, 1, \sqrt{2} + \epsilon)$. But $x_4$ is closer to the first representative, and so the resulting clustering places three points in the same cluster. As such, $\mathcal{C}$ is not found. $\qquad\square$

## Incremental clustering with extra clusters

**Lemma 5.1** Suppose $S$ has a nice $\ell$-clustering, for $\ell \leq k$. Then the points returned by CANDIDATES($S$) include at least one representative from each of these clusters.

*Proof.* Consider any nice $\ell$-clustering $\mathcal{C}$ of $S$. Single linkage will not join a point $x \in \mathcal{C}_i$ with $x' \in \mathcal{C}_j, j \neq i$ until $x$ is already connected to all the other points in $\mathcal{C}_i$. As a result, the single linkage tree will contain an internal node whose descendant leaves are exactly $\mathcal{C}_i$; and by construction, this node will be assigned a point in $\mathcal{C}_i$. Since this holds for all $i$, we see that there must be an $\ell$-pruning of the tree whose corresponding leaf-points induce $\mathcal{C}$. Finally, we note that any $\ell$-pruning of the tree consists of nodes at distance $< \ell$ from the root. $\qquad\square$

**Theorem 5.3** Suppose there is a nice $k$-clustering $\mathcal{C}$ of $\mathcal{X}$. Then for each $t$, the set $T_t$ has at most $2^{k-1}$ points, including at least one representative from each $C_i$ for which $C_i \cap \{x_1, \ldots, x_t\} \neq \emptyset$.

*Proof.* In what follows, let $S_t$ denote the first $t$ data points, $x_1, \ldots, x_t$.

We'll use induction on $t$; clearly it holds at $t = 0$.

Suppose it holds at time $t$. Since $T_t$ has a representative from each $\mathcal{C}_i$ that touches $S_t$, it is also true that $T_t \cup \{x_{t+1}\}$ has a representative from each $\mathcal{C}_i$ that touches $S_{t+1}$. Suppose there are $\ell$ such clusters $\mathcal{C}_i$. Then the corresponding sub-clusters $\mathcal{C}_i \cap (T_t \cup \{x_{t+1}\})$ are a nice $\ell$-clustering of $T_t \cup \{x_{t+1}\}$. By Lemma 5.1, applying CANDIDATES to this set will return a subset that still contains at least one representative of each of these clusters. $\qquad\square$

**Theorem 5.4** Pick any incremental clustering algorithm that maintains a list of $\ell$ centers that are guaranteed to be consistent with a target nice $k$-clustering. Then $\ell \geq 2^{k-1}$.

The construction involves points on a $(k-1)$-dimensional hypercube, under the $\ell_\infty$ metric. Pick any $a_1 > a_2 > \cdots > a_{k-1} > 0$ and consider the space of $2^{k-1}$ points

$$\mathcal{X} = \{-a_1, +a_1\} \times \{-a_2, +a_2\} \times \cdots \times \{-a_{k-1}, +a_{k-1}\}.$$

We will see that $(\mathcal{X}, \ell_\infty)$ has $2^{k-1}$ distinct nice $k$-clusterings, and that each individual point in $\mathcal{X}$ is a singleton cluster in at least one of these clusterings. Therefore, any $\ell$ points that are consistent with all nice $k$-clusterings must include each individual point, so $\ell \geq 2^{k-1}$.

It remains to characterize the nice $k$-clusterings. For any binary vector $b \in \{-1, +1\}^{k-1}$, consider the $k$-clustering $\mathcal{C}(b) = \{\mathcal{C}_1(b), \ldots, \mathcal{C}_k(b)\}$ defined as follows:

- $\mathcal{C}_1(b) = \{x \in \mathcal{X} : x_1 b_1 > 0\}$
- $\mathcal{C}_2(b) = \{x \in \mathcal{X} : x_1 b_1 < 0, x_2 b_2 > 0\}$
- $\mathcal{C}_i(b) = \{x \in \mathcal{X} : x_1 b_1 < 0, \ldots, x_{i-1} b_{i-1} < 0, x_i b_i > 0\}$ for $2 < i < k$
- $\mathcal{C}_k(b) = \{x \in \mathcal{X} : x_1 b_1 < 0, \ldots, x_{k-1} b_{k-1} < 0\}$

Notice that $\mathcal{C}_1(b)$ consists of all points whose first coordinate is $a_1 b_1$, while $\mathcal{C}_2(b)$ consists of all points whose first coordinate is $-a_1 b_1$ and whose second coordinate is $a_2 b_2$, and so on. We finish by showing that each $\mathcal{C}(b)$ is nice.

**Lemma 5.2** For any $b \in \{-1, +1\}^{k-1}$, the $k$-clustering $\mathcal{C}(b)$ is nice.

*Proof.* For any coordinate $1 \leq i < k-1$, the cluster $\mathcal{C}_i(b)$ consists of points that agree on the first $i$ coordinates. Therefore the maximum interpoint $\ell_\infty$ distance within this cluster is $2a_{i+1}$. Any other point in $\mathcal{X}$ disagrees with this cluster on at least one of these $i$ coordinates, and is thus at distance at least $2a_i$ from this cluster.

The last two clusters, $\mathcal{C}_{k-1}(b)$ and $\mathcal{C}_k(b)$, are singletons. $\qquad\square$

In this lower bound, the need for $2^{k-1}$ representatives stems from the non-uniqueness of the nice $k$-clustering. We conjecture that the bound holds even with uniqueness, and can perhaps be shown by suitably adapting the methodology of Theorem 3.8.

## Sequential $k$-means with extra clusters

**Theorem 5.6** Fix a data set $(\mathcal{X}, d)$ with a convex-nice clustering $\mathcal{C} = \{C_1, \ldots, C_k\}$ and let $\beta = \min_i |C_i|/|\mathcal{X}|$. If the points are ordered uniformly at random, then for any $\ell \geq k$, sequential $\ell$-means will return a refinement of $\mathcal{C}$ with probability at least $1 - ke^{-\beta\ell}$.

*Proof.* Let $\theta$ be the probability that the first $\ell$ points will include at least one point in each of the $k$ clusters of $\mathcal{C}$. Let $p_i$ be the probability of missing cluster $C_i$ after seeing $\ell$ points selected uniformly at random, so that

$$p_i \leq \left(1 - \frac{|C_i|}{|\mathcal{X}|}\right)^\ell \leq (1 - \beta)^\ell \leq e^{-\beta\ell}.$$

Then $\theta$ is greater than $1 - \sum_{i=1}^k p_i \geq 1 - ke^{-\beta\ell}$.

Assume this good event occurs, and the set of centers $T$ includes a representative from each cluster. Since $\mathcal{C}$ is convex-nice, every subsequent point will be assigned to a center within the convex hull of

its cluster, and that center will remain within the convex hull after it is updated. As a result, the final clustering produced by the algorithm is a refinement of $\mathcal{C}$. $\qquad\square$

**Theorem 5.7** Pick any $k \geq 3$. Consider any data set $\mathcal{X}$ in $\mathbb{R}$ (under the usual metric) that has a convex-nice $k$-clustering $\mathcal{C} = \{C_1, \ldots, C_k\}$. Then there exists an ordering of $\mathcal{X}$ under which sequential $\ell$-means with $\ell \leq \min_i |C_i|$ centers fails to return a refinement of $\mathcal{C}$.

*Proof.* Consider a data set $(\mathcal{X}, d)$ on the real line with a convex-nice clustering $\mathcal{C}$. Let $C_1$ be the leftmost cluster.

Now, consider an ordering of $\mathcal{X}$ that presents points from left to right. Then the initial $\ell$ centers all lie in $C_1$. Moreover, all the centers will continue to lie in the convex hull of $C_1$ while the points of $C_1$ are being processed.

Let $c$ be the rightmost center after all the points in $C_1$ are processed. The next point $x$ to appear lies to the right of $C_1$ and is thus assigned to center $c$, causing $c$ to move to the right, but not past $x$. Since points are processed left to right, this continues to hold for all remaining elements $x$: they are each assigned to $c$ and make $c$ move to the right, but not past $x$.

As such, all remaining elements only influence the position of center $c$, and leave the other centers unchanged within $C_1$. As a result, at most one of the final centers is outside the convex hull of $C_1$. Since there are at least three clusters in $\mathcal{C}$, this implies that the final clustering obtained by sequential $\ell$-means is not a refinement of $\mathcal{C}$. $\qquad\square$

## A broader class of clusterings

**Theorem 5.10** Consider any clustering $\mathcal{C} = \{C_1, \ldots, C_k\}$ of $(\mathcal{X}, d)$, with core $\{C_1^o, \ldots, C_k^o\}$. Let $\beta = \min_i |C_i^o|/|\mathcal{X}|$. Fix any $\ell \geq k$. Then, given any ordering of $\mathcal{X}$, Algorithm **??** detects a refinement of $\mathcal{C}$ with probability $1 - ke^{-\beta\ell}$.

*Proof.* By [1], Algorithm 5.9 selects $\ell$ points uniformly at random from the data. Now, let $\theta$ be the probability that the set of $\ell$ centers selected by Algorithm 5.9 includes at least one point from every cluster's core. By the same reasoning as in Theorem 5.6, we have $\theta \geq 1 - ke^{-\beta\ell}$.

Assume that the final set $T$ contains at least one center from each core. We argue that in that case, the clustering $\mathcal{C}'$ induced by $T$ is a refinement of $\mathcal{C}$. Consider point $x \in C_i$ for some $C_i \in \mathcal{C}$. Then, $x$ is closer to all elements in $C_i^o$ than to any element outside of $C_i$ and will thus be assigned to either a center in $C_i^o$ or some center in $C_i \backslash C_i^o$, but not to a point outside of $C_i$. $\qquad\square$