[Reviews · NeurIPS 2014]

Submitted by Assigned_Reviewer_19

Summary of the paper:

The paper studies the incremental clustering problem and shows several properties:
- It shows that no deterministic memory-bounded incremental clustering method is nice-detecting. Specifically, the authors show that no deterministic nice-detecting incremental clustering algorithm can use less than 2^{cp-1} bits of memory for data in R^p under the l2 metric.
- It shows that the sequential nearest-neighbour clustering is perfect detecting while the sequential k-means is not.
- It also shows that incremental algorithms can find a refinement of nice clustering if 2^{k-1} centers are allowed and this is a tight bound. Then some example algorithms are displayed.

General comments:

- The paper is written clearly and the guarantees in this paper are solid.
- The paper first shows that no memory-bounded incremental method is nice-detecting. This is a kind of “impossibility theorem”, which indicates the natural limitation of the incremental clustering algorithms. Although the result seems discouraging, we should note that detecting “nice” partitions may not be a necessary goal for incremental clustering algorithms since nice partitioning is a relatively strong assumption. But nevertheless, showing that the incremental clustering is strictly weaker than the batch model is also of great interest.
- The paper then shows that such limitation can be addressed by allowing more clusters and this is the crowning achievement in this paper. Although it is not the only paper that is aware of the benefits for extra clusters in incremental clustering (e.g., the streaming clustering algorithm proposed in [1] allows \Theta(k \log n) clusters in the loop and it can use only O(k \log n) memory to deliver an O(1) worst-case approximation.), this paper provides a tight bound on the number of extra clusters that is sufficient for a refinement of nice partitioning. This is also a satisfying contribution to incremental clustering.
- The paper seems to be a bit crowded, especially for the last part. I suggest the authors to reorganize the last several sections.

Minor typos:

- Line 568: each core contain… -> each core contains

[1] Shindler, M., Meyerson, A., and Wong, A. Fast and accurate k-means for large datasets. In NIPS, pages 2375-2383, 2011.
Summary: This is a well-written paper, and is a valuable contribution to the area of incremental clustering.

Submitted by Assigned_Reviewer_39

This paper presents a number of theoretical results characterizing the limitations/capabilities of incremental clustering. Below I summarize some of the key results, skipping some of less significance (from my point of view).

One of the key results is that there exists no deterministic memory-bounded incremental method that can guarantee to detect nice clustering (i.e., a clustering of data in which every element is closer to points in its own cluster than to points in other clusters) from data when a unique nice clustering exists. The main idea of the proof is to construct a data set with a unique nice clustering and demonstrat that any deterministic incremental algorithm that's memory bounded will necessarily fail with some ordering of the data.
The paper also shows that perfect clustering, a stronger notion of clusterability, can be detected by a single-link based incremental algorithm, but not by the popular incremental k-means.
By allowing extra clusters, the paper shows that an incremental algorithm can find a refinement of the target nice clustering if allowed 2^{k-1} centers and also shows that it is not possible to use fewer cluster centers.

Given that the definition of nice clustering requires a somewhat global view of the clustering structure, it is somewhat intuitive that memory bounded incremental algorithms will fail. Non-the-less, the lower bound results are quite interesting and original (to the best of my knowledge). I followed most of the proofs provided in the paper as well as the appendix and they appear to be sound and some of them non-trivial. I think they present solid contribution to the theoretical understanding of incremental clustering.

The presentation of the paper has some room for improvement. In particular, the paper itself lacks balance in its space allocation. The first result took 6 pages and was given the (almost) full proof, whereas the later results are provided without any proof or even sketch of proof. A reader will have to read the supplementary materials to get through the rest of the paper. Further more, the supplementary materials were not separated from the main paper. Rather, an extended version was presented as the supplementary. This makes it difficult to find the proof for a particular theorem because the numbers are not consistent. I appreciate the attempt to provide the high level intuition for the proof of the first result, but I find the explanation difficult to understand without the detailed proof. I think this explanation needs some work to make it really useful (perhaps with some figures?). Another issue with the presentation is that it seems to be trying to fit too much into the paper and some of the results seem dis-connected from the rest of the paper. For example, the convex-nice seems to be mainly introduced to help sequential k-means? All previous sections focus on deterministic algorithms, whereas section 5 jumps to a random algorithm, which again seems only loosely connected.

Another thing on my wish list for this paper is some concrete practical understanding of these results. For example, the lower bound result on using extra clusters is quite discouraging, requiring at least 2^{k-1} clusters in order to find a proper refinement of the target nice-clustering. In practice, will this be what we typically observe? I understand that this is a theoretical paper, none-the-less, it would have been nice to see some (perhaps) synthetic experiments to demonstrate what to expect in reality when the proposed algorithm is put to use.
Summary: Overall, I think this paper makes some solid contributions to the theoretical understanding of incremental clustering. The presentation of the paper needs some work but I think is doable.

Submitted by Assigned_Reviewer_44

This paper provides a set of new theoretical results of incremental clustering.

Quality:
Although there is no experimental analysis, results are technically sound and discussions are carefully addressed.
Clarity:
The paper is clearly written overall and easy to understand. But there is no conclusion, please add it.
Originality:
Results seem to be new and original.
Significance:
Although this paper is well written and interesting, my main concern is that main results without extra clusters are about only two extreme settings. That is, Theorem 3.8 merely says that there exists no universal algorithm that can learn "any" nice clustering, while Theorem 4.3 says that the class of perfect clusterings is learnable but the perfectness is restrictive and is violated in many typical scenarios. So the significance is not very high as it stands and it would be very interesting if some intermediate learnable class, which is practically reasonable, is found to be learnable in the incremental setting.

Other comments:
- Related to the above concern, is there any advantage in sequential k-means (Algorithm 2.2) compared to the sequential nearest-neighbor clustering (Algorithm 2.4)? That is, is there a class of clusterings that sequential k-means can learn while the sequential NN clustering cannot?
- Some concrete values are used in Definition 3.4. Isn't it possible to remove such values from the definition of M-configuration?

Pros:
- New theoretical results
- Careful theoretical analysis of incremental clustering
- Good presentation
Cons:
- The main result is provided for only extreme setting
Summary: A well written paper with new theoretical results about incremental clustering, but the results are not impressive.

Submitted by Assigned_Reviewer_46

The paper looks at situations where incremental clustering might do as well as "batch" clustering. After a series of results establishing the weakness of incremental clustering, the paper presents an interesting result where an incremental algorithm, if allowed to use more clusters, can produce a clustering that refines the true answer. The authors also identify a special case of clusterings (ones with "large cores") where a simple sampling-based replacement strategy will also recover the correct answer.

I'm very torn about this paper. While it makes a good effort in pursuit of the goal (understanding incremental clusterings), it's hardly surprising that incremental strategies can fail miserably even when batch algorithms do well. Having said that, the 4 point example is nice (although I'd recommend the authors look at the streaming MEB paper by Chan and Zarrabi-Zadeh from CCCG a few years ago to find that examples constructed in a similar vein).

The observation about finding refinements is a good one. While I might not like 2^k clusters instead of k, it's a neat idea to even make this connection and be able to prove a result. I'm less excited about the result on finding cores: that seems to follow from the fact that the cores are reasonably large.

This paper is very well written and extremely clear. So much so that I must commend the authors: it's often tempting to cram as much material as one can at the expense of clarity, and the authors do an excellent job of both laying out the main results AND giving intuition for them within the rather onerous NIPS page limits.
Summary: An interesting study of incremental approaches to clustering
Author Feedback
Author rebuttal: Reviewer 1:

Thank you very much for your review, the nice comments on our paper, and the reference to the work of Shindler, Meyerson, and Wong. We will incorporate your suggestions and reorganize the latter parts of the paper.

Reviewer 2:

Thank you very much for your review and the nice comments on our paper. We will fix the supplementary material so that the numbers are consistent with the main paper and incorporate your suggestions for the main paper.

Reviewer 3:

Thank you very much for your review.

The advantage of sequential k-means (and other variants of k-means) for some applications can be shown using the weighted clustering framework [2]. Intuitively, sequential k-means is more sensitive to the density of the data than methods such as sequential nearest-neighbor.

The concrete values in Definition 3.4 are given only to make the proof of the main result easier to follow.